# The Influence of Social Support on Leisure-Time Physical Activity of the Elderly in the Chinese Village of Fuwen

**DOI:** 10.3390/healthcare11152193

**Published:** 2023-08-03

**Authors:** Jiayi Zhou, Chen Yang, Jiabin Yu, Xiaoguang Zhao, Jinan Wu, Zhiyong Liu, Jianshe Li, Yaodong Gu

**Affiliations:** 1Faculty of Sport Science, Research Academy of Grand Health, Ningbo University, Ningbo 315211, China; nbuzhoujiayi@outlook.com (J.Z.);; 2Department of Physical Medicine and Rehabilitation, Northwestern University, 710 N Lake Shore Dr, Chicago, IL 60611, USA; cyang03@ricres.org; 3Max Nader Lab for Rehabilitation Technologies and Outcomes Research, Shirley Ryan Ability Lab, 355 E Erie St., Chicago, IL 60611, USA

**Keywords:** elderly, recreational physical activity, family support, rural population, motivators, China

## Abstract

The purpose of this study was to examine the associations of social support factors with leisure-time physical activity (LTPA) of older people in Fuwen village. A cross-sectional study included 523 randomly selected elderly people (60+ years) whose LTPA levels were determined using the shortened version of the International Physical Activity Questionnaire (IPAQ-S). A modified version of the Physical Activity Social Support Scale (PASSS) was operated to gather perceived scores of the social support factors. A multivariate linear regression was performed to locate associations of perceived scores of social supports with leisure-time walking (LTW) and moderate and vigorous physical activity (MVPA). The results indicated that social support from family was positively and significantly related to LTW and MVPA in both models. The community factor was positively and significantly correlated with MVPA in both models. The sport club factor was related to LTW and MVPA to some extent. The results suggest that social support from family is the most important motivator for older people’s LTW and MVPA in the village of Fuwen. Social support from the community is the motivator for older people’s MVPA. The sport club factor has some effects on older people’s LTW and MVPA as well. More future studies are needed to extend the database of the relationship between social support and rural older people’s physical activity.

## 1. Introduction

Physical activity is important to maintain physical and mental health for all people. There is plenty of evidence supporting the effectiveness of regular physical activity in the primary and secondary prevention of chronic diseases, including cardiovascular disease, diabetes, depression, premature death, and so on [1]. Leisure-time physical activity (LTPA) refers to body movements that involve energy expenditure caused by skeletal muscle contraction in leisure time, and consists of LTW and MVPA [2]. The World Health Organization suggested that the guideline of MVPA for adults to obtain various health benefits is 150 min per week. However, a systematic review study found that health benefits can be achieved even at physical activity volumes that are half or less than that of the current guideline [3]. Population aging is a common social problem related to the increasing medical burden on many countries, and this problem is even more serious in the countryside in China. According to the China Aging Research Report 2022, the percentage of older people aged over 60 years old in the countryside was 7.99% higher than that in the cities [4]. Considering the health benefits acquired by older people from physical activity [5,6], encouraging older people to take part in more physical activity, particularly LTPA, is vital for preserving their health and reducing medical costs.

According to the socio-ecological model, internal factors, social support factors and built environmental factors are the main influential factors for physical activity [7]. These three factors have been found to be related to physical activity levels by numerous studies [8,9,10,11,12,13,14,15,16,17]. For example, Yarmohammdi et al. reviewed 34 studies and suggested that health status is an internal factor that was highlighted in most articles, either as a motivator (5 studies) or a barrier (18 studies) for older people. Additionally, health improvement and health benefit were other common motivators. Fear of walking outside, fear of falling, and lack of time were mentioned as barriers of older people’s physical activity in internal factors [18]. For environmental factors, Yun et al. reviewed 70 papers and found that walkability, urbanization, land use mix-diversity, accessibility, walking amenities, and bicycle lanes were positive environmental factors for older people’s physical activity [19].

The association between social support factors and physical activity was also investigated in many previous studies. These studies have focused on various participants groups, including patients [20,21,22,23,24,25,26,27,28], youths [10,29,30,31], adults [32], and a limited number of studies specifically focused on older people [12,33,34,35]. For patients, positive social support from healthcare providers played an important role in promoting their physical activity [24]. For youth, parental social support was demonstrated to have a positive effect on youth’s physical activity. The effect of social support on physical activity may alter due to participant differences. For example, social support from staff and peers is the important motivator for older people’s physical activity, but social support from family is the vital motivator for youths [36]. Even fewer studies were specifically focused on rural older people [37,38]. Marthammuthu et al. found that with an increase in the social support score, rural-dwelling older women were more likely to have higher physical activity (odds ratios =1.22; 95% confidence interval [1.10, 1.34]; *p* < 0.001) [38].

There were some studies that investigated the influence factors of physical activity in China, and most of them focused on the environmental factors [39,40,41,42,43]. The significant environmental factors may differ due to differences in the cities, age groups, and sex groups in China. For example, Yu et al. found that the significant environmental factors of older people’s physical activity differed between the cities of Hangzhou and Wenzhou. Residential density is only significantly related to physical activity level in Wenzhou, and access to services is only significantly correlated with physical activity in Hangzhou [44]. Sun et al. found that street connectivity, walking/cycling facilities, and aesthetics were significantly associated with men’s LTPA, but no significant environmental factors were found to be related to women’s LTPA in the city of Xi’an [40]. Wu et al. found street pavement slope and street connectivity were significantly related to older people’s LTPA in the city of Nanjing [45], but Yu et al. found that high perceived esthetic was associated with LTPA of adults aged 18–69 years old [42].

Compared with numerous studies about environmental factors, few studies focused on internal and social support factors. Yikeranmu et al. found that both self-efficacy and social support are positively related to physical activity of adolescents aged 18–21 years old in the city of Xi’an [46]. To our knowledge, no previous study specifically focused on the effect of social support on rural older people’s physical activity in China. Therefore, the purpose of this study was to locate the association of social support factors with older people’s physical activity level in the village of Fuwen, and furthermore to discuss how to motivate older people to engage in more physical activity from the viewpoint of modifying social support.

## 2. Materials and Methods

This study was conducted in the village of Fuwen, Hangzhou, Zhejiang Province in the eastern region of China. The official resident population of Fuwen village was 8377 at the end of 2019. According to our survey, the regular physical activities of residents included the following: walking around the village, square dancing in the open place, and taking some exercise using public fitness equipment, and so on. A cross-sectional survey of random samples was performed in May 2023. The demographic characteristics of 523 older people as well as their LTW, MVPA, and perceived scores of social supports were collected by two team members in the way of face-to-face interviews. The interviewers were familiar with the questionnaires and the process of interview before performing the interviews. The village committee assisted us to organize older people taking part in the interviews. The inclusion criteria were the following: (a) over 60 years old, (b) resident of Fuwen village, (c) no cognitive impairment, and being able to communicate and consent. This study was approved by the ethics committee of the Research Academy of Grand Health, Ningbo University (RAGH20230500012; May 2023).

The questionnaire used in this study includes three parts of demographic characteristics, LTPA, and social support. Demographic variables involved gender, age, height, weight, lower extremity musculoskeletal disorders, self-reported health status. Lower extremity musculoskeletal disorders refer to whether the participants have lower extremity musculoskeletal disorders to affect them engaging in physical activity. There are three levels in the self-report health status, including excellent, good, and bad. The body mass index (BMI) was computed using body height and weight. IPAQ-S was used to collect the raw data of LTW, moderate-intensity physical activity, and vigorous-intensity physical activity in the last seven days. The metabolic equivalent scores (METs) of LTW and MVPA were computed after the interview based on the IPAQ scoring procedure. The MET values of walking, moderate-intensity physical activity, and vigorous-intensity physical activity are 3.3, 4, and 8, respectively. Since sitting does not have a MET value according to the IPAQ scoring procedure and it is not a type of physical activity, it was not reported in this study. We modified the IPAQ-S into three questions. Q1: during the last 7 days, how much time did you usually spend walking lasting over 10 min. Answer: __days per week, __hours per day. Q2: during the last 7 days, how much time did you usually spend doing moderate physical activities, like fast walking, square dancing, table tennis, badminton, cycling? Answer: __days per week, __hours per day. Q3: during the last 7 days, how much time did you usually spend doing vigorous physical activities like running, fast cycling, fast rope skipping? Answer: __days per week, __hours per day.

The social support part was designed with reference to the Physical Activity Social Support Scale (PASSS) used in previous studies [12,32]. All of the questions regarding social support part involved 3-point Likert Scales (1—totally disagree, 2—partly agree, 3—totally agree). A higher score means a better social support received by older people. Except for family and friend factors included in the PASSS, government, sport club, and community factors were added in the social support part of this study. There were three questions in the family element. Q1: invite me to take exercise or walking; Q2: encourage me to take exercise or walking; Q3: join me to take exercise or walking. The questions in the friend element were same as the family element. The government element included three questions. Q1: put up posters about exercise; Q2: update or maintain pubic sports facilities; Q3: organize sport activities. Three questions were included in the sport club element. Q1: organize activities that you like to take part in; Q2: encourage older people to engage in the organized activities; Q3: organized activities were suitable for older people to participate. Four questions were involved in the community element. Q1: there were plenty of exercise facilities. Q2: encourage you to engage in the organized activities; Q3: there were places for older people’s exercise. Q4: sport instruct volunteer assist you to take exercise. The perceived score of each element of social support is the average score of the included questions. The Cronbach α coefficient of the modified version of the PASSS was 0.977, which proved that the social support questionnaire used in this study was reliable.

Descriptive statistical analyses were used to describe demographic variables, LTW, MVPA, LTPA and perceived scores of the social supports. A multivariate linear regression was performed to locate the association of perceived scores of social supports with LTW and MVPA. Two models were conducted in the regression analyses. Model 1 only included social support factors, and Model 2 included both social support factors and demographic variables. The statistical significance level was set as *p* < 0.05. All of the statistical analyses were carried out in SPSS 19.0 software (IBM Inc., Chicago, IL, USA).

## 3. Results

Table 1 shows the demographic characteristics of the participants in this study. The percentages of male and female participants were almost equal. The largest number of respondents were aged 60–69 years (67.3%), and the average age of the participants was 66.6 years. The BMI values of most participants were 18–24 kg/m^2^, and the mean BMI of all of the participants was 23.2 kg/m^2^. Of the participants, 70.2% did not report having lower extremity musculoskeletal disorders. Concerning health status, 76.9% of the participants thought they had a good health status. Very few participants felt they were in excellent health, with only 3.2%. Another 19.9% of the participants considered they were in bad health.

Table 2 shows the descriptive results of older people’s LTW, MVPA, LTPA, and perceived social support scores in the village of Fuwen. LTW was the main way of LTPA for older people of this study, with the average level of 874.08 MET minutes per week. Compared with LTW, the MVPA level of older people was quite low, with an average level of just 597.71 MET minutes per week. Considering the higher MET value of MVPA than LTW in the IPAQ scoring procedure, the time spent on MVPA by older people should be much less than the time spent on LTW. Regarding the perceived social support scores, the family score was the highest, and the friends score was a little lower. The scores of the sport club and community elements were the lowest.

Table 3 shows the results of the correlation analysis of perceived social support scores with LTW, MVPA, and LTPA.

Table 4 reveals the association results between the perceived social support scores and older people’s LTW in two different models. Without adjustments with the demographic information of the participants, the family and sport club scores were significantly related to older people’s LTW, and both of them were positively related to LTW. With adjustments of the demographic information of the participants, the explanation degree of the regression model increased little by 2.5%, and only the family score was still positively associated with older people’s LTW at a significant level. Additionally, the age, lower extremity musculoskeletal disorders and self-reported health status scores were significantly related with LTW. Except for lower extremity musculoskeletal disorders, both the age and self-reported health status scores were negatively associated with LTW. For other social support elements, friends, government, and community were not significantly related to older people’s LTW in both of the models. For other demographic variables, the gender, BMI, and self-reported health status scores were not significantly correlated with older people’s LTW.

Table 5 displays the associations between the perceived social support scores and older people’s MVPA in the two models. Without adjusting for demographic information, the family, sport club and community values were significantly related to the MVPA. Except for the sport club element, both the family and community elements were positively associated with MVPA. After adjusting by demographic information, the explanation degree of the regression model increased by 5.7%, and the association of social support factors with older people’s MVPA did not change. For the demographic factors, BMI and health status were significantly related to MVPA, with a positive relationship for BMI and a negative relationship for health status. For other social support elements, friends, government, and community were not significantly related to older people’s MVPA. For other demographic variables, gender, age, and lower extremity musculoskeletal disorders were not significantly associated with older people’s MVPA.

Table 6 displays the associations between the perceived social support scores and older people’s LTPA in the two models. Without adjusting for demographic information, the family, sport club and community scores were significantly related to the LTPA. Except for the sport club element, both the family and community elements were positively associated with LTPA. After adjusting for demographic information, the explanation degree of the regression model increased by 5%, and the friends element was found to be negatively associated with LTPA. The significant associations of family, sport club, and community with LTPA did not change. For demographic factors, BMI and health status were significantly related to LTPA, with a positive relationship for BMI and a negative relationship for health status.

## 4. Discussion

The purpose of this study was to examine the association of social support with older people’s LTW and MVPA, and to discuss how the social support factors affect older people’s LTPA. Finally, the results of this study may provide some insights to increase older people’s LTPA in the village.

Our results indicate that support from family is the most important social factor for older people’s LTPA in the village of Fuwen. In the association analysis results, the perceived score for the family element was positively related to older people’s LTW and MVPA at significant levels (Table 4 and Table 5), which means the more family support received by older people, the more LTW and MVPA older people engaged in. The positive effect of family on older people’s LTPA did not alter whether or not the social support factors were adjusted according to demographic variables. The perceived score of the family element was also the highest among all of the social support factors (Table 2), which means support from family is the most vital element for older people in the village of Fuwen. This result is consistent with findings of previous studies. In a systematic review study, Spiteri et al. synthesized fifty-five previous studies and discovered that social support from family was the key factor for older people’s physical activity. A lack of support from family would hinder older people from taking part in physical activity [17]. In another population-based study, Boehm et al. found that older people who had the company of family or friends to walk had a 2.45 times higher prevalence to reach the recommendation of at least 150 min LTW and MVPA per week than those who did not [12].

Support from the community is a motivator for older people’s MVPA, but not for older people’s LTW. Our results showed that the perceived score of community was positively associated with older people’s MVPA at a significant level, and the association was not affected by the models used (Table 5). No significant association was found between the perceived scores of community and older people’s LTW (Table 4). This result indicates that community support is important for older people’s MVPA, but not for LTW, and this result is explicable. Questions in the community element in this study included the perceived conditions of exercise equipment, exercise places, and the number of sports instruction volunteers. These conditions are important when older people engage in MVPA. More exercise equipment, exercise places, and sports instruction volunteers are helpful for older people taking part in more MVPA. However, LTW does not always require these conditions. Walkable roads, clear air, and beautiful environments may be more vital for older people engaging in LTW. These results are in line with those of previous studies. Zheng et al. found that a sense of community was associated significantly with older people’s MVPA [47].

For other social factors, the sport club element was found to be related to LTW and MVPA. It was positively related to LTW, but the positive association disappeared after the model was adjusted with demographic variables. The positive association of sport club with LTW was consistent with the results of a previous study [48]. Surprisingly, the sport club element was negatively correlated with MVPA in both regression models, which was opposite to its positive association with LTW. One possible explanation is that sports clubs in the village of Fuwen organize activities that more related to LTW, such as brisk walking, which is popular in China. The friends and governments elements were not found to have a significant association with LTW and MVPA in both models, and the friend element was negatively related to LTPA in the adjusted model. This result was not in line with those of previous studies. Quite a few previous studies suggested that support from friends was an important motivator for residents’ LTPA [15,49,50,51,52,53]. Loprinzi et al. discovered that compared with those with 0 close friends, older people with 1 to 2, 3 to 4, and 5 to more than 6 close friends had 1.70-, 2.38-, 2.57-, and 2.71-fold increased odds of reaching physical activity guidelines [54]. A possible reason for the current contradiction may be that the older participants of this study were from a village. Compared with older people living in the city, older people in the village may have contact with friends less frequently because of the lower residential density. Compared with friend support, family support is easier to obtain for older people living in villages.

For the demographic variables, age and lower extremity musculoskeletal disorders elements were significantly associated with older people’s LTW. The participants who were younger did not have lower extremity musculoskeletal disorders, and took part in more LTW (Table 4). The BMI and self-reported health status elements were significantly related to older people’s MVPA. Older people with higher BMIs and better health status engaged in more MVPA (Table 5). Additionally, we noticed that the LTPA level of older people in the village of Fuwen was much lower than for older people living in the city. In our previous studies, the average LTPA level of older people in the city of Hangzhou was 2048 MET minutes per week compared with 1470 MET minutes per week of this study [44]. The LTPA levels of older people living in other cities from China were also much higher than that of this study [41,42,55,56]. Both the LTW and MVPA levels were at a low level for older people in the village, and especially for MVPA, at only 597.71 MET minutes per week. Whether this phenomenon exists in other villages requires more future studies. In any case, increasing older people’s LTPA levels in all possible ways is an urgent and vital need, considering the significant health benefits of LTPA that older people could obtain from it [57,58].

There are some limitations in this study. Firstly, bias may have inevitably been included in the data due to the self-reported interview used in this study. Secondly, demographic variables like living alone or with a partner, whether living with descendants or not, and whether having domestic animals were not included in this study. Not all possible demographic variables affecting older people’s LTPA were included in this study, and this may have had some effects on the regression results. Thirdly, whether the results of this study could be extended to older people living in other villages in China and in other countries needs more future studies.

## 5. Conclusions

Social support from family was the most important motivator for older people’s LTW and MVPA who live in the village of Fuwen. Social support from community was the motivator for older people’s MVPA. The sport club factor had some effect on older people’s LTW and MVPA, but more studies are needed to confirm this. The demographic variables including age, lower extremity musculoskeletal disorders, BMI, and health status affected older people’s LTW or MVPA to some extent. For policy makers, enhancing family support is the most important route to encourage older people in the village to take part in more LTPA. Moreover, the low level of older people’s LTPA in the village needs to be brought to the forefront for government, and improved.

## Figures and Tables

**Table 1 healthcare-11-02193-t001:** Demographic characteristics of the participants in the village of Fuwen (*n* = 523).

Variable	*n*	*%*	Mean ± SD	95% CI
Gender				
Men	264	50.5		
Women	259	49.5		
Age			66.6 ± 9.0	60–88
60–69 years	352	67.3		
70–79 years	112	21.4		
≥80 years	59	11.3		
BMI			23.2 ± 2.9	18–29.2
<18	8	1.7		
18–24	325	62.5		
>24	183	35.8		
Lower extremity musculoskeletal disorders				
Yes	156	29.8		
No	367	70.2		
Self-reported health status				
Excellent	17	3.2		
Good	402	76.9		
Bad	104	19.9		

Note: BMI means body mass index, and the units are kg/m^2^; SD means standard deviation. 95% CI indicates the 95% confidence interval values.

**Table 2 healthcare-11-02193-t002:** Description of older people’s LTW, MVPA, LTPA, and perceived social support scores in the village of Fuwen (*n* = 523).

Variable.	Mean ± SD	95% CI
LTW (MET minutes per week)	874.08 ± 828.47	(0, 2772)
MVPA (MET minutes per week)	597.71 ± 1088.17	(0, 3360)
LTPA (MET minutes per week)	1470.86 ± 1494.18	(0, 6426)
Family	2.65 ± 0.50	(1.70, 3.00)
Friends	2.61 ± 0.52	(1.67, 3.00)
Government	2.59 ± 0.58	(1.00, 3.00)
Sport club	2.44 ± 0.61	(1.00, 3.00)
Community	2.44 ± 0.67	(1.00, 3.00)

Note: MET represents metabolic equivalent score, SD means standard deviation. 95% CI indicates the 95% confidence interval values.

**Table 3 healthcare-11-02193-t003:** Results of correlation analysis of perceived social support scores with LTW, MVPA, and LTPA in the village of Fuwen (*n* = 523).

Variable	LTW Coefficient	*p*	MVPA Coefficient	*p*	LTPA Coefficient	*p*
Family	0.30	<0.001 *	0.14	<0.001 *	0.27	<0.001 *
Friends	0.24	<0.001 *	0.09	0.03 *	0.20	<0.001 *
Government	0.19	<0.001 *	0.12	0.01 *	0.19	<0.001 *
Sport club	0.20	<0.001 *	0.04	0.37	0.14	0.001 *
Community	0.14	0.001 *	0.23	<0.001 *	0.25	<0.001 *

Note: Coefficient represents Pearson correlation coefficient, * represents significant correlation (*p* < 0.05).

**Table 4 healthcare-11-02193-t004:** Associations between the perceived social support scores and older people’s LTW in two models.

Variable	*B*	Model 1 *SE*	Beta	*p*	*B*	Model 2 *SE*	Beta	*p*
Family	533.82	143.47	1.20	<0.001 *	636.04	145.61	1.43	<0.001 *
Friends	−183.59	143.86	−0.41	0.20	−241.27	141.32	−0.53	0.09
Government	−130.90	117.60	−0.29	0.27	−89.00	115.04	−0.20	0.44
Sport club	211.21	97.50	0.44	0.03 *	196.96	96.49	0.41	0.42
Community	−92.16	92.00	−0.19	0.32	−112.17	89.45	−0.24	0.21
Gender					−88.27	67.06	−0.12	0.19
Age					−7.87	3.65	−0.44	0.03 *
BMI					13.89	9.84	0.27	0.159
Lower extremity musculoskeletal disorders					220.70	90.28	0.32	0.02 *
Self-reported health status					−94.94	92.93	−0.18	0.31

Note: Dependent variable: total score of older people LTW. B stands for regression coefficient, SE represents standard error, Beta means standardized beta coefficients, * represents significant association (*p* < 0.05). Model 1 represents the regression model of social support factors; the F value of model 1 was 137.04, *p* < 0.001; the adjusted R^2^ of model 1 was 0.565. Model 2 represents the regression model of social support factors adjusted with demographic characteristics; the F value of model 2 was 76.25, *p* < 0.001; the adjusted R^2^ of model 2 was 0.590.

**Table 5 healthcare-11-02193-t005:** Associations between the perceived social support scores and older people’s MVPA in two models.

Variable	*B*	Model 1 *SE*	Beta	*p*	*B*	Model 2 *SE*	Beta	*p*
Family	463.45	186.50	1.01	0.01 *	502.34	186.67	1.09	0.007 *
Friends	−265.32	187.01	−0.57	0.16	−274.43	181.18	−0.59	0.13
Government	−191.10	152.87	−0.41	0.21	−135.25	147.49	−0.29	0.36
Sport club	−568.66	126.74	−1.15	<0.001 *	−585.00	123.71	−1.19	<0.001 *
Community	799.40	119.60	1.63	<0.001 *	773.44	114.67	1.58	<0.001 *
Gender					0.54	85.97	0.001	0.99
Age					−1.24	4.68	−0.07	0.79
BMI					58.04	12.62	1.09	<0.001 *
Lower extremity musculoskeletal disorders					−77.17	115.74	−0.11	0.51
Self-reported health status					−584.76	119.14	−1.04	<0.001 *

Note: Dependent variable: total score of older people’s MVPA. B stands for regression coefficient, SE represents standard error, Beta means standardized beta coefficients, * represents significant association (*p* < 0.05). Model 1 represents the regression model of social support factors; the F value of model 1 was 47.65, *p* < 0.001; the adjusted R^2^ of model 1 was 0.308. Model 2 represents the regression model of social support factors adjusted with demographic characteristics; the F value of model 2 was 31.12, *p* < 0.001; the adjusted R^2^ of model 2 was 0.365.

**Table 6 healthcare-11-02193-t006:** Associations between the perceived social support scores and older people’s LTPA in two models.

Variable	*B*	Model 1 *SE*	Beta	*p*	*B*	Model 2 *SE*	Beta	*p*
Family	997.61	256.70	1.28	<0.001 *	1138.95	253.76	1.47	<0.001 *
Friends	−449.17	257.41	−0.57	0.08	−515.16	246.29	−0.65	0.04 *
Government	−321.47	210.42	−0.41	0.13	−222.78	200.49	−0.28	0.27
Sport club	−357.99	174.45	−0.43	0.04 *	−388.04	168.16	−0.47	0.02 *
Community	706.83	164.62	0.85	<0.001 *	660.81	155.88	0.80	<0.001 *
Gender					−89.99	116.87	−0.07	0.44
Age					−8.88	6.37	−0.29	0.16
BMI					71.83	17.16	0.80	<0.001 *
Lower extremity musculoskeletal disorders					142.37	157.33	0.12	0.37
Self-reported health status					−686.35	161.95	−0.73	<0.001 *

Note: Dependent variable: total score of older people’s MVPA. B stands for regression coefficient, SE represents stand error, Beta means standardized beta coefficients, * represents significant association (*p* < 0.05). Model 1 represents the regression model of social support factors; the F value of model 1 was 124.20, *p* < 0.001; the adjusted R^2^ of model 1 was 0.54. Model 2 represents the regression model of social support factors adjusted with demographic characteristics; the F value of model 2 was 75.98, *p* < 0.001; the adjusted R^2^ of model 2 was 0.59.

## Data Availability

Data are available on request due to privacy restrictions. The data presented in this study may be available on request from the corresponding author.

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
