# Peer review of "The Influence of Social Support on Leisure-Time Physical Activity of the Elderly in the Chinese Village of Fuwen"

_healthcare, 2023, doi:10.3390/healthcare11152193_

Round 1
Reviewer 1 Report
In this study, the authors examined the influence of social support (family, community and sports clubs) on the physical activity of people over 60 years old in a village in China. The data the authors analyzed were collected by filling in the IPAS-Q questionnaire and a modified version of the PASSS scale, to estimate types of social support. Physical activity of the subjects was categorized into leisure time walking (LTW) and moderate to vigorous physical activity (MVPA). The results showed that each of the three types of social support is associated with at least one type of physical activity, and the best result was found for family support, which is positively associated with both LTW and MVPA.
I have some objections:
The text must be proofread by an English teacher or a native speaker. It cannot be written that the purpose of the work was to "locate" the association (line 12), that the results are significant in "both two models" (lines 20 and 21), etc.
Title – if you start the title with „how“, then there should be a question mark at the end. Still, I suggest that you change the title from a question form to a statement, like „The influence of social support on leisure-time physical activity of the elderly in the Chinese village of Fuwen“
Line 6 – there is a semicolon extra
Lines 12-13 – please change „to locate“ with „to examine“, rephrase the second part of this sentence into „with leisure-time physical activity (LTPA) of elderly people in Fuwen village.“
Lines 13-15 – please combine these two sentences into one „A cross-sectional study included 523 randomly selected elderly people (60+ years) whose LTPA level was determined using the shortened version of the International Physical Activity Questionnaire (IPAQ-S).“
Keywords – please change the word „rural“ to term „rural population“ and add „China“
line 30 – physical activity is important for all people, not only adults
lines 33-35 – you wrote „Leisure time physical activity (LTPA) is consisted of LTW and MVPA, and would bring more health benefits for adults due to larger energy consumption.“ – Why „and“ in front of „would“? More health benefits compared to what?
line 37 – please add „… can be achieved even at ….“
line 40 - when you write "in the Chinese village", it means that there is a specific village called a Chinese village where you did your research. And here you want to say that there are old people in the countryside in China.
line 41 - here you wrote that more old people live in the countryside than in the city, but you did not specify the age limit for old age
lines 42-43 - You wrote here that older people will have greater health benefits from physical activity than young people. First, the papers you cited do not claim that, and second, I do not know why you think so?
line 48 – there cannot be „abundant studies2, but can be many studies or numerous studies
lines 50-21 – if you use „either“, then there should also be „or“
line 68 – „order women“?
line 70 – change to „… factors of physical activities in China, …“
Materials & Methods
You should better describe the village where you conducted the research - how many people live there, what proportion of the 60+ inhabitants of that village are your
respondents, what opportunities do the residents have for engaging in physical activity? Do your respondents live alone or with a partner? Do any of them live with descendants? Do their family members live near them or do they encourage elderly people to be active by phone?
Considering that you respondents live in the countryside, do they have domestic animals that they need to take care of? Because people who have domestic animals should be physically active and take care of these animals even if no one encourages them.
You did not explain why you chose age 60 as the age limit for elderly - is that the age people retire in China? If I look from my position, where in Europe people work at least until the age of 65, there is a possibility that your sample consists of both retirees and working people. If so, then you cannot analyze the sample as a homogeneous group when part of the people have to work at least 8 hours a day, maybe some office work.
How did you get the data on the subjects' BMI - did you measure the subjects or did they just enter their height and weight in the questionnaire?
line 83 – who are the „interview members“?
line 90 – change „included“ to „includes“
line 98 – change „Likert scaled“ to „Likert scales“
lines 122-123 – change to „ … percentage of male and female participants was …“
Lines 123-124 – Please change to „The largest number of respondents were aged 60-69 years (67.3%), and the average age of all participants was 66.6 years.“
I am sorry but I really don't have time to correct sentence by sentence, the manuscript needs to be reviewed by a native speaker.
Table 1 - Does any of your respondents have a BMI>28 kg/m2 (is anyone obese)?
Table 4 – please correct the last digit in the "p" column, it moves to the next row
line 186 – delete „s“ in „factors“
Once again, please have someone who knows English grammar correct the manuscript.
The text must be proofread by an English teacher or a native speaker.
Author Response
We would like to thank the reviewer 1 for the careful analysis he/she has done of our manuscript and the positive evaluation of this paper. His/her comments contributed to a significant improvement to our manuscript. According to the suggestion, we revised the manuscript according.
Point 1:
The text must be proofread by an English teacher or a native speaker. It cannot be written that the purpose of the work was to "locate" the association (line 12), that the results are significant in "both two models" (lines 20 and 21), etc.
Title – if you start the title with „how“, then there should be a question mark at the end. Still, I suggest that you change the title from a question form to a statement, like „The influence of social support on leisure-time physical activity of the elderly in the Chinese village of Fuwen“
Line 6 – there is a semicolon extra
Lines 12-13 – please change „to locate“ with „to examine“, rephrase the second part of this sentence into „with leisure-time physical activity (LTPA) of elderly people in Fuwen village.“
Lines 13-15 – please combine these two sentences into one „A cross-sectional study included 523 randomly selected elderly people (60+ years) whose LTPA level was determined using the shortened version of the International Physical Activity Questionnaire (IPAQ-S).“
Keywords – please change the word „rural“ to term „rural population“ and add „China“
line 30 – physical activity is important for all people, not only adults
Response: Thank you for the suggestions and the time you spent on the review of this paper. We have changed the title and revised those words you mentioned.
Point 2:
lines 33-35 – you wrote „Leisure time physical activity (LTPA) is consisted of LTW and MVPA, and would bring more health benefits for adults due to larger energy consumption.“ – Why „and“ in front of „would“? More health benefits compared to what?
Response: We have modified this sentence into “Leisure time physical activity (LTPA) refers to body movements that involve energy expenditure caused by skeletal muscle contraction in leisure time, and is consisted of LTW and MVPA.” Please see lines 36-38
Point 3:
line 37 – please add „… can be achieved even at ….“
line 40 - when you write "in the Chinese village", it means that there is a specific village called a Chinese village where you did your research. And here you want to say that there are old people in the countryside in China.
line 41 - here you wrote that more old people live in the countryside than in the city, but you did not specify the age limit for old age
Response: We have modified them according to your suggestion.
Point 4:
lines 42-43 - You wrote here that older people will have greater health benefits from physical activity than young people. First, the papers you cited do not claim that, and second, I do not know why you think so?
Response: We have revised this sentence into “Considering the health benefits acquired by older people from physical activity [5, 6], encouraging older people to take part in more physical activity, particularly LTPA, is vital for preserving their health and reducing medical cost.” Please lines 45-48
Point 5:
line 48 – there cannot be „abundant studies2, but can be many studies or numerous studies
lines 50-21 – if you use „either“, then there should also be „or“
line 68 – „order women“?
line 70 – change to „… factors of physical activities in China, …“
Response: We have modified them according to your suggestion.
Point 6:
You should better describe the village where you conducted the research - how many people live there, what proportion of the 60+ inhabitants of that village are your respondents, what opportunities do the residents have for engaging in physical activity?
Response: We searched the possible official information about Fuwen village, and the latest official data only shows the population of Fuwen at the end of 2019, and no more detail like proportion of 60+ inhabitants. According to our survey, the physical activities of residents include walking around the village, square dancing in the open place, and taking some exercise using public fitness equipment, and so on. We have added the above information in the method section. Please see lines 97-100
Point 7:
Do your respondents live alone or with a partner? Do any of them live with descendants? Do their family members live near them or do they encourage elderly people to be active by phone?
Considering that you respondents live in the countryside, do they have domestic animals that they need to take care of? Because people who have domestic animals should be physically active and take care of these animals even if no one encourages them.
Response: We absolutely agreed with your opinion that those factors you mentioned might make some effect on elderly’s physical activity. However, the physical activity level might be affected by many factors, not limited to the factors involved in this study (demographic variables and social support factors) and the factors you mentioned. According to the socio-ecological model, internal factors, and built environment factors are also the influential factors for physical activity. We don’t think all possible factors could be included in one study. The information you mentioned are not included in the PASSS and the demographic questionnaire, so examining the effects of these factors is unfortunately not the focus of the current study. We added the factors you mentioned in the limitation section “Secondly, the demographic variables like living alone or with a partner, whether living with descendants, whether having domestic animals were not included in this study. Not all possible demographic variables affecting older people’s LTPA were included in this study, and this might have some effects on the regression results.” Please see lines 294-297
Point 8:
You did not explain why you chose age 60 as the age limit for elderly - is that the age people retire in China? If I look from my position, where in Europe people work at least until the age of 65, there is a possibility that your sample consists of both retirees and working people. If so, then you cannot analyze the sample as a homogeneous group when part of the people have to work at least 8 hours a day, maybe some office work.
Response: According to results of the Seventh National Population Census published by Chinese government in 2021, the proportion of people aged over 60 years old were 18.7%, and people aged over 60 were regarded as elderly in China. Therefore, we chose 60 as the age limit for elderly.
Point 9:
How did you get the data on the subjects' BMI - did you measure the subjects or did they just enter their height and weight in the questionnaire?
Response: Demographic variables of this study involved gender, age, height, weight, motion sickness, self-report health status. Height and weight were self-reported. Body mass index (BMI) was computed using body height and weight.
Point 10:
line 83 – who are the „interview members“?
Response: The raw data were collected by two team members. We have changed the “interview members” to “interviewers “. Hope it works.
Point 11:
line 90 – change „included“ to „includes“
line 98 – change „Likert scaled“ to „Likert scales“
lines 122-123 – change to „ … percentage of male and female participants was …“
Lines 123-124 – Please change to „The largest number of respondents were aged 60-69 years (67.3%), and the average age of all participants was 66.6 years.“
Response: We have modified them according to your suggestions.
Point 12:
Table 1 - Does any of your respondents have a BMI>28 kg/m2 (is anyone obese)?
Response: yes, we have added the 95%CI in table 1. Please lines 166-167
Point 13:
Table 4 – please correct the last digit in the "p" column, it moves to the next row
line 186 – delete „s“ in „factors“
Response: We have modified them according to your suggestions.
Reviewer 2 Report
See attached

Minor editing is needed. See comments in general and line recommendations.
Author Response
We would like to thank the reviewer 2 for the careful analysis he/she has done of our manuscript and the positive evaluation of this paper. His/her comments contributed to a significant improvement to our manuscript. According to the suggestion, we revised the manuscript according.
Point 1:
Leisure Time Physical Activity (LTPA) should be expressed for what it is, MET minutes per week. I am unsure why you have separated the walking domain but combined the moderate and vigorous domain?
It would be best to look at total MET minutes and then separate them all afer the fact (walking, moderate, vigorous). Also, siƫng tme is not mentioned, which is an important variable that should be included.
Response:
Thank you for the suggestion. We have modified the unit of LTPA.
Because the purpose of this study was to investigate the effect of social support on physical activity, and sitting is not physical activity and doesn’t have a MET value according to IPAQ scoring procedure, so we didn’t mention sitting time. We separated LTPA into LTW and MVPA according to previous studies[1-3].
Point 2:
Why motion sickness??? Is this an interpretation/translation error? I am unsure why that is included in the demographic info?
Response:
What we want to express using “Motion sickness” is that do the participants have diseases of lower extremity motor system to affect them taking exercise. We have changed the word into “Lower extremity motor dysfunctions” in the whole manuscript.
Point 2: A random sample? I am curious where the random sample was taken? Is it in a gym, shopping center, grocery store, etc.? It maters because depending upon where the random sample was collected may predetermine your “random” sample… is it random gym goers or a true reflection of the general population.
Response:
The survey was assisted by the village committee because we have a co-operative project with the village committee of Fuwen. The participants of this study were randomly recruited by the village committee. There are not gym or shopping center in the Chinese village.
Point 3:
97-113 The PASS is explained in detail, but you don’t do the same for the other questionnaires (demographic, IPAQ) … I would recommend that you either omit or modify the over explanation of the PASS and include more explanation regarding the other two questionnaires.
Response:
According to your suggestion, we have added more explanation of the other two questionaries. Please lines 112-116, 119-129.
we explained the details of PASSS because we enlarged the PASSS questionaries. Except family and friend, we added government, sport club, and community factors in this study.
Point 4:
115-16 you state “The multivariate linear regression was performed to locate the association of perceived scores of social supports with LTW and MVPA.” This to me suggests that you should first find the associations (correlation table) which also helps us further specify and define our dependent variables to be used in a multiple regression analysis... if needed.
Response:
We agreed with your opinion, and also think first finding the associations, and then using multiple regression analysis is a right way to analyse the association relationship. However, in numerous previous studies, multiple regression analysis is directly used without correlation analysis [2, 4-8]. Therefore, we think the method used in this study is also suitable for investigating the association relationship of social support and physical activity.
Point 5:
117 Why two models? Is there a specific reason for this? Please explain.
Response:
Model 1 was used to examine the association of social support with LTW and MVPA. Model2 was used to confirm that whether the association relationship changed after adjusted with demographic variables. The results showed that the association of social support with LTW changed after adjusted with demographic variables (Table 3). Sport club factor wasn’t related with LTW any more after adjustment. the association of social support with MVPA was not affected after adjustment. The similar operation could be found in previous studies[1, 5, 9].
Point 6:
See other comments. Is this a representation of the general population (equal males/females) for the age range/community?
Response: We are not sure whether equal males/females is a representation of general population. According to the available official information, the percentage of male and female are almost equal (50.4 vs 49.6) at the year of 2011. The official resident population of Fuwen village is 8,377 at the end of 2019.
Point 7:
Table 2 I am unfamiliar with PASS but is this how it is normally scored or utilized…separation of the individual score?
Response: PASSS is 3-point Likert Scale. A higher score means a better social support received by older people. The perceived score of each element of social support is the average score of included questions. We have added the score explanation in the method section. Please see lines 145-146.
Point 8:
Table 3 Standardized Beta scores should be included 0.575, 0.85, etc. See above comment, why two models? Please explain.
Response:
We are sorry that we don’t understand the meaning of “Standardized Beta scores should be included 0.575, 0.85, etc.” About why two models, please see point 5.
Point 9:
160/175 Do you mean dependent variable (rather than depend variable)?
Response:
We have revised this word according to your suggestion.
Point 10:
If I am not mistaken, 7 independent variables are used, and you have a N of over 500 participants… this falls outside the recommended ratio 20-40 subjects for every independent variable. In doing so, you may have significant correlation even though the correlation may not exist.
Response: Including the demographic variables, 10 independent variables are included in the regression model. The similar and even larger size of subjects could be found in previous studies[1, 10-12].
Point 11:
Also, you have gathered those data from one questionnaire which leads me to believe (I am not familiar with the PASS) that most of those variables are already correlated with each other (Similar to your IPAQ data)… As additional predictor variables are included in the model when the predictor variables are correlated, the precision of the computed regression coefficients diminishes. I don’t believe that multiple regression analysis is the correct statistical model to use (from the start). It would be best if you first provided a simple correlation table and then (maybe) provide a multiple regression as a secondary analysis (only one model with relevant data from the correlation tables).
Response: The similar study design (demographic variables, social support or built environment or internal factors, LTPA) was widely used to investigate the affect factors of physical activity in previous studies. You can find numerous studies in system review study, like the reference[13]. About the regression method used in this study, please see point 4. We absolutely agreed with your opinion about the method, but considering the similar regression method was also used in previous studies, we think the method used in this study is also suitable. Hope you can understand.
Point 12:
You don’t mention anything regarding outliers and if outliers were controlled for, or how you handled outliers. I would also argue that nothing is mentioned about how this sample of older adults is a true representation of the population of China/rural/older adult population in general
Response: The significant factors of physical activity might be different in different cities and countries in previous studies, so we are not confident that the results of this study could be the representation of the population in China. We announced it in the third point of limitation “Thirdly, whether the results of this study could be extended for older people living in other villages in China and other countries needs more future studies.” To our knowledge, very few studies investigate the association of social support with older people’s physical activity in a Chinese village, so this paper make sense.
About outlier, we controlled it in the interview process. 3 participants self-reported super high LTPA that should not be real, we deleted the data of those 3 participants directly. We have added 95% CI in table 1 and table 2.
Reference
- Boehm, A. W.; Mielke, G. I.; da Cruz, M. F.; Ramires, V. V.; Wehrmeister, F. C., Social Support and Leisure-Time Physical Activity Among the Elderly: A Population-Based Study. Journal of Physical Activity & Health 2016, 13, (6), 599-605.
- Yu, T.; Fu, M.; Zhang, B.; Feng, L.; Meng, H.; Li, X.; Su, S.; Dun, Q.; Cheng, S.; Nian, Y.; Wu, Q.; Meng, Z.; Duan, Y.; Liu, X.; Chen, L.; Wu, N.; Zou, Y., Neighbourhood built environment and leisure-time physical activity: A cross-sectional study in southern China. Eur. J. Sport Sci. 2020, 1-8.
- Zhu, W.; Chi, A.; Sun, Y., Physical activity among older Chinese adults living in urban and rural areas: A review. J. Sport. Health. Sci. 2016, 5, (3), 281-286.
- Van Cauwenberg, J.; Clarys, P.; De Bourdeaudhuij, I.; Van Holle, V.; Verte, D.; De Witte, N.; De Donder, L.; Buffel, T.; Dury, S.; Deforche, B., Physical environmental factors related to walking and cycling in older adults: the Belgian aging studies. BMC Public Health 2012, 12, 142.
- Van Dyck, D.; Cerin, E.; Conway, T. L.; De Bourdeaudhuij, I.; Owen, N.; Kerr, J.; Cardon, G.; Frank, L. D.; Saelens, B. E.; Sallis, J. F., Perceived neighborhood environmental attributes associated with adults' leisure-time physical activity: Findings from Belgium, Australia and the USA. Health & Place 2013, 19, 59-68.
- Wu, Z. J.; Song, Y. L.; Wang, H. L.; Zhang, F.; Li, F. H.; Wang, Z. Y., Influence of the built environment of Nanjing's Urban Community on the leisure physical activity of the elderly: an empirical study. Bmc Public Health 2019, 19, (1), 11.
- Yu, J.; Yang, C.; Zhang, S.; Zhai, D.; Li, J., Comparison Study of Perceived Neighborhood-Built Environment and Elderly Leisure-Time Physical Activity between Hangzhou and Wenzhou, China. Int. J. Environ. Res. Public Health 2020, 17, (24).
- Siu, V. W.; Lambert, W. E.; Fu, R.; Hillier, T. A.; Bosworth, M.; Michael, Y. L., Built environment and its influences on walking among older women: use of standardized geographic units to define urban forms. J. Environ. Pub. Health 2012, 2012, 203141-203141.
- Yu, T.; Fu, M. Z.; Zhang, B. Z.; Feng, L. J.; Meng, H. D.; Li, X.; Su, S. L.; Dun, Q. Q.; Cheng, S. Y.; Nian, Y. P.; Wu, Q. X.; Meng, Z. Q.; Duan, Y. T.; Liu, X.; Chen, L. W.; Wu, N. J.; Zou, Y. L., Neighbourhood built environment and leisure-time physical activity: A cross-sectional study in southern China. European Journal of Sport Science.
- Sun, Y.; He, C.; Zhang, X.; Zhu, W., Association of Built Environment with Physical Activity and Physical Fitness in Men and Women Living inside the City Wall of Xi'an, China. Int. J. Environ. Res. Public Health 2020, 17, (14).
- Zhou, R.; Li, Y.; Umezaki, M.; Ding, Y.; Jiang, H.; Comber, A.; Fu, H., Association between physical activity and neighborhood environment among middle-aged adults in Shanghai. J. Environ. Pub. Health 2013, 2013, 239595-239595.
- Sugiyama, T.; Cerin, E.; Owen, N.; Oyeyemi, A. L.; Conway, T. L.; Van Dyck, D.; Schipperijn, J.; Macfarlane, D. J.; Salvo, D.; Reis, R. S.; Mitas, J.; Sarmiento, O. L.; Davey, R.; Schofield, G.; Orzanco-Garralda, R.; Sallis, J. F., Perceived neighbourhood environmental attributes associated with adults' recreational walking: IPEN Adult study in 12 countries. Health Place 2014, 28, 22-30.
- Barnett, D. W.; Barnett, A.; Nathan, A.; Van Cauwenberg, J.; Cerin, E.; Grp, C. O. A. W., Built environmental correlates of older adults' total physical activity and walking: a systematic review and meta-analysis. Int. J. Behav. Nutr. Phy. 2017, 14.
Round 2
Reviewer 1 Report
The manuscript is substantially improved, I recommend it to be accepted for publication.
Author Response
Dear reviewer,
Thank you for the help of improvement of this manuscript.
Jiabin
Reviewer 2 Report
Please see attached document

Just minor corrections.
Author Response
Response to Reviewer 2 Comments
We would like to thank the reviewer 2 for the careful analysis he/she has done of our manuscript and the positive evaluation of this paper again.
Point 1:
I would argue that the IPAQ siƫtng time calculations (Physical inactivity) is just as good (if not better) at detecting social support. Its one simple variable that if the IPAQ was completed would be very simple to include in your data.
Just because previous studies have done something does not mean it is the correct way you need to do it. My argument would be that you are trying to determine which variables are most assocated with social support, total METs is possibly your best predictor correct? However, taking that a step further, which of the three total MET contributors (light, moderate, vigourous) is most associated with or influenced by social support. Response: Thank you for the suggestions and the time you spent on the review of this paper. We have changed the title and revised those words you mentioned.
Response: According to your suggestions, we have added table 6 to show the association relationship between perceived social support scores and LTPA (total MET). About sitting time, we agree with your opinion about the importance of it for detecting social support. We are so sorry that we didn’t collect this data because we thought it is not a type of physical activity at the beginning of this study. We will collect sitting time in future study. Thank you for the suggestion.
Point 2:
Motion sickness – Lower extremity motor dysfunction… I am sorry I don’t understand! I believe this is a translation error but maybe a more in depth explanation would suffice. Is it a mobility score that has been validated?
Response: we have changed Lower extremity motor dysfunction into “Lower extremity musculoskeletal disorders”. What we want to express is that whether the participants have lower extremity musculoskeletal disorders to affect them engaging in physical activity. The participants answered yes or no in the interview. If this translation also doesn’t make sense, could you give us some suggestions on the translation of this question?
Point 3:
See my comment above… in general comments. Please read the assumtions section in particular. https://statistics.laerd.com/spss-tutorials/multiple-regression-using-spss-statistics.php Here you can also find what I mean by standarized beta scores.
Response: According to your suggestions, we have added table 2 to show the results of correlation analysis of perceived social support scores with LTW, MVPA, LTPA, respectively.
Point 4:
See point 4 comments and link
Response: We have added standardized Beta coefficients in table 4, 5, 6